# Effects of Bioactive Peptides from Atlantic Salmon Processing By-Products on Oxyntopeptic and Enteroendocrine Cells of the Gastric Mucosa of European Seabass and Gilthead Seabream

**DOI:** 10.3390/ani13193020

**Published:** 2023-09-26

**Authors:** Paolo Clavenzani, Giulia Lattanzio, Alessio Bonaldo, Luca Parma, Serena Busti, Åge Oterhals, Odd Helge Romarheim, Tone Aspevik, Pier Paolo Gatta, Maurizio Mazzoni

**Affiliations:** 1Department of Veterinary Medical Sciences, *Alma Mater Studiorum*—University of Bologna, Via Tolara di Sopra 50, 40064 Ozzano Emilia, Italy; paolo.clavenzani@unibo.it (P.C.); giulia.lattanzio@studio.unibo.it (G.L.); alessio.bonaldo@unibo.it (A.B.); luca.parma@unibo.it (L.P.); serena.busti2@unibo.it (S.B.); pierpaolo.gatta@unibo.it (P.P.G.); 2Nofima, the Norwegian Institute of Food Fisheries and Aquaculture Research, 5141 Fyllingsdalen, Norway; aage.oterhals@nofima.no (Å.O.); odd.h.romarheim@nofima.no (O.H.R.); tone.aspevik@nofima.no (T.A.)

**Keywords:** gilthead seabream and European seabass, bioactive peptide, oxyntopeptic cells, neuropeptide Y, somatostatin and ghrelin

## Abstract

**Simple Summary:**

Processing fish or by-products, as well as the so-called bycatch of fishing, generates a considerable amount of waste. The by-products of fish (or marine invertebrates) processing can be an interesting source of nutrients with high nutritional value. They can be reused as functional ingredients in the feed industry. Hydrolysates have been used as chemoattractants and fishmeal replacers in aquafeed, due to their low molecular weight compounds and balanced amino acid profiles. Peptides with predicted anti-inflammatory, immunostimulatory/anti-microbial properties were identified in the different fractions of the by-products using state-of-the-art peptidomics and bioinformatics techniques (often referred to as the in silico approach).

**Abstract:**

The present study was designed to evaluate the effects of dietary levels of bioactive peptides (BPs) derived from salmon processing by-products on the presence and distribution of peptic cells (oxyntopeptic cells, OPs) and enteric endocrine cells (EECs) that contain GHR, NPY and SOM in the gastric mucosa of European seabass and gilthead seabream. In this study, 27 seabass and 27 seabreams were divided into three experimental groups: a control group (CTR) fed a control diet and two groups fed different levels of BP to replace fishmeal: 5% BP (BP5%) and 10% BP (BP10%). The stomach of each fish was sampled and processed for immunohistochemistry. Some SOM, NPY and GHR-IR cells exhibited alternating “open type” and “closed type” EECs morphologies. The BP10% group (16.8 ± 7.5) showed an increase in the number of NPY-IR cells compared to CTR (CTR 8.5 ± 4.8) and BP5% (BP10% vs. CTR *p* ≤ 0.01; BP10% vs. BP5% *p* ≤ 0.05) in the seabream gastric mucosa. In addition, in seabream gastric tissue, SOM-IR cells in the BP 10% diet (16.8 ± 3.5) were different from those in CTR (12.5 ± 5) (CTR vs. BP 10% *p* ≤ 0.05) and BP 5% (12.9 ± 2.5) (BP 5% vs. BP 10% *p* ≤ 0.01). EEC SOM-IR cells increased at 10% BP (5.3 ± 0.7) compared to 5% BP (4.4 ± 0.8) (5% BP vs. 10% BP *p* ≤ 0.05) in seabass. The results obtained may provide a good basis for a better understanding of the potential of salmon BPs as feed ingredients for seabass and seabream.

## 1. Introduction

The fishing industry has grown steadily over the last decade. This growth has been accompanied by a high volume of protein-rich by-products. This waste includes whole or parts of fish such as fillets, skin and fins, bones, heads, guts and scales. In the context of a circular economy, these by-products are rich in proteins, which can be recovered/reused as functional ingredients in the feed industry. Recently, several studies have reported the utilization of fish by-products by enzymatic hydrolysis for the recovery of various valuable components, and fish protein hydrolysates, such as capelin, mackerel, hoki frame, jumbo squid, yellow stripe trevally and tuna liver, have been shown to possess antioxidant activity with the ability to scavenge hydroxyl radicals, superoxide anion radicals, hydrogen peroxide and chelate metal ions [1,2,3,4,5,6]. Fishery proteins represent a potential source of biopeptides.

Previous studies have described the biological activity of marine protein hydrolysates produced from different species [7,8,9]. Protein hydrolysates are composed of low molecular weight compounds with a balanced amino acid profile. These characteristics have stimulated research and several studies have reported that these hydrolysates are interesting chemotactic agents and can be used as fishmeal substitutes in aquatic feeds [10,11,12,13,14].

The protein hydrolysates obtained contain peptides. To identify which of these peptides have anti-inflammatory, immunostimulatory or antimicrobial properties, state-of-the-art peptidomic and bioinformatic techniques (often referred to as the in silico approach) were applied. In this context, interest in the in silico approach has increased because it is less costly and time-consuming [15,16].

With its high nutritional value and potential pharmacological applications, farmed Atlantic salmon is a popular food around the world. The fileting operation generates large amounts of by-products consisting of head, backbone, skin and viscera. Mechanically separated salmon muscle is readily available as a low-cost by-product of the filleting process and can account for up to 60% (*w*/*w*) of filleting waste [17]. Several studies have evaluated the biochemical functional properties of salmon muscle protein hydrolysate (and its respective peptides) [18,19,20] and there is growing industrial interest in the utilization of bioactive peptides (BPs) within the fish feed and pet food industry. Recently, in European seabass, the dietary inclusion levels of bioactive peptides from farmed Atlantic salmon have shown the possibility of an almost total replacement of fish meal in a plant-based diet in terms of growth and feed efficiency [21].

The regulation of food intake relies heavily on the gut–brain axis. Several studies have highlighted the important role played by gut hormones in response to food digestion [22,23]. These hormones are involved in appetite regulation as short-term peripheral satiety signals. They promote satiety, i.e., a decrease in appetite and a reduction in food intake, through endocrine and nervous pathways by activating various signaling pathways [24,25,26,27]. In vertebrates, appetite and digestion are controlled by the enteroendocrine system [28,29]. Gut–brain hormones, such as ghrelin (GHR), neuropeptide Y (NPY), somatostatin (SOM), cholecystokinin, etc., are important factors in the control of feeding behaviours [28,29]. GHR, NPY and SOM have been found in a large number of fish species. Their tissue distribution supports the idea that GHR has an integrative role in the regulation of energy balance at both the central nervous system and systemic levels [30]. NPY and GHR are reported to correlate positively with feed intake [28]. GHR, a small peptide hormone secreted by the stomach, is an appetite stimulator [31]. GHR levels increase before a meal and decrease postprandially [32] and are involved in the regulation of appetite, energy balance and body weight [33]. NPY is a potent, highly conserved, multi-functional peptide found in vertebrates, including fish. It plays an important role in the regulation of feeding behavior, energy metabolism and digestive processes [34,35,36,37]. In fish, SOMs have many direct and indirect effects on intermediary metabolism and feeding behavior. In general, SOMs inhibit food intake and promote catabolic processes (e.g., mobilizing stored lipids and carbohydrates) [36,38].

Certain hormone-like peptides obtained by protein hydrolysis have the potential to affect gastrointestinal (GI) motility, endocrine metabolism, intake and animal performance [39]. Most bioactive peptides (BPs) share some structural features. These features are represented by the length of the peptide residue (from 2 to 20 amino acids) [40], the presence of proline, lysine or arginine, and the presence of hydrophobic amino acids [41]. Regulatory peptides with hormone-like activity (hormone peptides) or the ability to modulate blood levels of certain hormones could also be obtained by enzymatic hydrolysis. The biological activity of hormone-like peptides is typically mediated by their interaction with G protein-coupled receptors (GPCRs) on the cell surface and further activation of the ligand–receptor signaling pathway to regulate various physiological functions of the body [42]. Fish proteins may also serve as a reservoir of hormone-like peptides. Interestingly, some authors have found neuropeptide immuno-related molecules and molecules capable of binding to specific hormone receptors on cell membranes in fish protein hydrolysates [43,44,45]. In this study, we investigated the effect of supplementing salmon by-products derived from enzymatic hydrolysis on the presence and distribution of oxytocic (OP) and enteroendocrine (EEC) cell subpopulations expressing GHR, NPY and SOM in the gastric mucosa of European seabass and seabream.

## 2. Materials and Methods

### 2.1. Protein Hydrolysate

Atlantic salmon (*Salmo salar*) processing waste consisting of fresh heads and backbones was purchased from Biomega (Øygarden, Norway). The material was then blended in tap water (1:1) heated to 50 °C, to which chymotrypsin (0.1% *w*/*w*; Enzyme Supplies, Oxford, UK) was added. Hydrolysis was performed for 60 min, followed by deactivation of enzyme activity at T > 90 °C for 10 min. The hydrolyzate obtained (in the aqueous phase) was subsequently separated from the lipid and bone phases and dried to a dry powder by means of a NIRO P-6.3 spray dryer (GEA, Skanderborg, Denmark). The inlet and outlet temperatures were 200 and 92 °C, respectively.

### 2.2. Feed Production

The diets were formulated with FM and with a mixture of vegetable ingredients that are currently used in aquafeed for European seabass and gilthead seabream [46,47]. The control diet (CTR) was formulated to resemble a commercial feed for European seabass (*Dicentrarchus labrax*) and gilthead seabream (*Sparus aurata*) (Table 1). Two experimental feed mixes were formulated with an exchange of the fishmeal with the experimental fish protein hydrolysate at 5% (BP5) and 10% (BP10) levels, respectively [21]. The composition of the fish protein hydrolysate is reported in Table 2. Diets were preconditioned in an atmospheric double differential preconditioner (Wenger Manufacturing Inc., Sabetha, KS, USA) prior to extrusion on a TX-52 twin screw extruder (Wenger) and expanded through 2.5 mm dies to 3.2 mm pellets. The pellets were dried in a hot-air double-layer carousel dryer (model 200.2, Paul Klockner GmbH, Nistertal, Germany) at a constant air temperature and were coated with oil in a vacuum to achieve the final lipid content. Diets were formulated with FM and a mixture of vegetable ingredients currently used in aquafeed for European seabass and seabream [46,47].

### 2.3. Chemical Analysis

The nitrogen content was analyzed by the Kjeldahl method (ISO 5983-2, 2009 [49]) and the crude protein content was estimated on the basis of N × 6.25. The ash was determined by combustion of the raw material at 550 °C (ISO 5984-2, 2002 [50]). The dry matter content was determined by drying at 103 °C (ISO 6469-2, 2002 [51]). The fat content of the protein hydrolysate was analyzed by the EU method (Commission Directive 98/64/EC), while the fat content of the raw material was analyzed based on ethyl acetate extraction (NS 9402). Peptide size distributions were measured by HPLC size exclusion chromatography (SEC) (1260 series HPLC Agilent Technologies, Santa Clara, CA, USA) using a Superdex Peptide 10/300GL column (GE Healthcare, Uppsala, Sweden). The eluent was acetonitrile with TFA and UV detection at 190–600 nm [52].

### 2.4. Rearing Condition and Sampling

Juvenile European seabass (*Dicentrarchus labrax*) and gilthead seabream (*Sparus aurata*) were collected from an Italian commercial hatchery and reared in recirculating aquaculture systems (RAS) at the Aquaculture Laboratory of Cesenatico, Department of Veterinary Sciences, University of Bologna, Italy. At the beginning of the experimental procedures, sixty seabass and sixty seabreams were individually weighted and allocated, based on the species, in each of eighteen conical-bottomed tanks with a volume of 800 L connected to an RAS supplied with natural seawater (overall water volume: 22 m^3^; oxygen level 8.0 ± 1.0 mg L^−1^; temperature 24 ± 0.5 °C, salinity 28–32 g L^−1^) according to Busti et al. [53]. Over a period of 58 days, each experimental diet was administered twice daily until full satiety using an overfeeding approach employed for both species as described in Parma et al. [46,47]. At the end of the trial, at 12 h after a meal, three fish per tank were randomly selected from the 120 fish (60 gilthead seabream and 60 seabass) used for the performance studies, giving a total of 27 seabass (mean weight 147.1 g) and 27 gilthead seabreams (mean weight 168.76 g). Subsequently, the previously selected fish were sacrificed under anesthesia (excess of anesthetic MS222, 300 mg L^−1^). The GI tract from the esophagus to the posterior intestine was gently removed from each seabass and seabream and isolated from the coelomic cavity. The stomachs were isolated and fixed in formalin (pH 7.2) for 48 h at room temperature. After fixation, the stomachs were divided symmetrically by cutting along the long axis to obtain two equal halves. The stomach samples were dehydrated in a graded alcohol series, cleared in xylene and paraffin embedded. Sections (6 µm thick) placed on polylysine slides were obtained from each block of paraffin.

The Ethical-Scientific Committee for Animal Experimentation of the University of Bologna, Italy (ID 113/2020-PR) evaluated and approved all experimental procedures.

### 2.5. Feed Intake and Growth Calculation

The calculations used to determine the various performance parameters were as follows.
Specific growth rate (SGR) (% day − 1) = 100 ∗ (ln FBW − ln IBW)/days (where FBW and IBW are the final and initial body weights, respectively);
Feed intake (FI) (g kg ABW-1 day-1) = ((100 ∗ total intake)/(ABW))/days) (where average body weight, ABW = (IBW + FBW)/2);
Feed conversion ratio (FCR) = feed intake/weight gain

### 2.6. Immunohistochemistry

The stomach sections were processed for double labeling immunofluorescence. Table 3 lists the primary and secondary antibodies used in this study. Sections were deparaffinized, rehydrated and incubated with appropriate normal serum (5% normal goat or donkey serum) and 1% BSA diluted in PBS (phosphate buffer saline 0.01 M pH 7.4) for 1 h at room temperature (RT) to reduce non-specific secondary antibody binding. The sections were then incubated for 48 h at 4 °C in a humidity chamber with the following primary antibodies: rabbit anti Na^+^/K^+^-ATPase 1:600, rat anti SOM 1:500, mouse anti GHR 1:500 and goat anti NPY 1:1000. After washing, the sections were incubated for 1 h at RT with goat anti-rabbit AlexaFluor 594 at 1:1000, donkey anti-goat AlexaFluor 488 at 1:1200, donkey anti-mouse AlexaFluor 594 at 1:1200, or goat anti-rat at 1:1000. Finally, the sections were coverslipped with buffered glycerol, pH 8.6. After performing some preliminary tests, since we observed that some immunoreactive NPY cells coexpressed GHR, anti-NPY and anti-GHR antibodies were used simultaneously to evaluate the EECs that colocalized NPY/GHR, while the rat antiserum anti-SOM was used together with Na^+^K^+^-ATPase antibody, to characterize the distribution of OPs which express SOM.

### 2.7. Threshold Binarization Procedure

The following method was used to characterize the area occupied by the OPs immunoreactive (-IR) cells in the gastric mucosa, which was previously described by Mazzoni et al. [50]. Briefly, after scanning the preparations with a digital camera (Nikon DS-Qi1Nc, Nikon Instruments Europe BV, Amsterdam, The Netherlands ) using the 20× objective and the NIS Elements BR 4.20.01 software (Nikon Instruments Europe BV, Amsterdam, The Netherlands), binarization was performed in a selected region of the gastric mucosa. This method allows the evaluation of the area occupied by the OPs-IR cells within the considered area. Gastric morphometric evaluations were performed by two investigators in a blinded fashion.

### 2.8. Antibody Specificity

Specificity for Na^+^K^+^-ATPase, GHR and SOM antibodies was previously demonstrated by Mazzoni et al. [54]. The specificity of the NPY antibodies was demonstrated by the absence of immunostaining when the antibodies were preabsorbed with an excess of the homologous peptide. Omitting the primary antibody prevented the secondary antibody from binding.

### 2.9. Morphometric Evaluations and Statistical Analysis

The specimens were examined using a Nikon Eclipse Ni microscope and images were taken using a Nikon DS-Qi1Nc digital camera and NIS Elements software BR 4.20.01 (Nikon Instruments Europe BV, Amsterdam, The Netherlands). Minor adjustments to contrast and brightness were made using Corel Photo Paint, while figure panels were prepared using Corel Draw (Corel Photo Paint and Corel Draw, Ottawa, ON, Canada). The 20× objective was used for morphometric evaluation. In the gastric mucosa, the area occupied by OPs-IR in 4.1 mm^2^ (0.410 × 10 fields) was measured by binarization (described above). In addition, the number of GHR, NPY and SOM IRs in 4.1 mm^2^ were counted in the gastric mucosa.

For each experimental group (CTR, BP 5% and BP 10%), the values obtained for OPs-IR area and the number of EECs were corporate and the means were calculated. The results were expressed as mean ± standard deviation (SD). The data were analyzed by one-way ANOVA (Graph Prism 4, GraphPad Software version 4.01, Inc., La Jolla, CA, USA). The experimental group was considered as the main effect. In addition, the means were then separated using the Tukey-HSD test. A *p* ≤ 0.05 was considered statistically significant.

## 3. Results

### 3.1. Protein Hydrolysate

The obtained spray-dried protein hydrolysate contained 82.8% protein, of which 82.5% was water-soluble (Table 2). The fat level was comparable to normal fishmeal; however, the ash level was lower due to the removal of bones by centrifugation after the hydrolysis process. No attempt was made to reduce the fat level further; however, an additional separation step has the potential to increase the protein level above 90%. The molecular weight distribution of the soluble protein fraction showed minor parts (5.2%) above 4 kDa and a high fraction (54%) below 1 kDa (Table 2).

### 3.2. Feed Intake and Growth

In European seabass, no significant differences among treatments were detected con-cerning final body weight, FBW (range 145.1–149.2 g), feed intake, FI (range 1.47–1.53% bw), specific growth rate, SGR (range 1.18–1.23% day^−1^) and FCR (range 1.30–1.31). Simi-larly, no differences among treatments were observed in gilthead seabream for FBW (166.7–170.0 g), FI (1.84–1.87% bw), SGR (1.43–1.46% day^−1^) and FCR (1.36–1.39).

### 3.3. Seabream and Seabass Morphological Features

The gastric mucosa of the seabass as well as that of the seabream is lined with prismatic epithelium composed of poorly stained cells with a nucleus in a basal position. These epithelial cells, probably responsible for the production of mucin, line the lumen of the stomach; these cells perform a protective function by interposing themselves between the gastric glands and the contents of the stomach. Below the epithelial layer, simple tubular gastric glands (in some cases bifid) have been observed over the entire surface of the stomach. Below the latter, a robust muscularis mucosae delimits the passage between the mucosa and submucosa.

Immunofluorescence with the Na^+^/K^+^-ATPase antibody confirmed the presence of OPs in all parts of the stomach: indeed, they showed intense immunoreactivity and appeared to be distributed along the simple tubular gastric glands (Figure 1B,C). However, the immunoreactivity of OPs is abruptly disrupted in the transition zone between the gastric and esophageal mucosa as well as at the gastrointestinal junction.

EECs-IR have been observed mainly above the glandular adenomas, while a smaller number of these cells are found mixed with OPs. Generally, EECs intermingled between the mucous cells tend to reach the endoluminal side (Figure 2 and Figure 3). These EECs located in correspondence with the endoluminal surface show an “open type” morphological aspect (“open type” EECs). These endocrine elements have an elongated or pyriform shape with two cytoplasmic extensions: one extension tends to reach the endoluminal side, insinuating itself between the epithelial cells, while the other, moving in the opposite direction, reaches the basement membrane. Other EECs exhibit “closed type” morphological features (“closed type” EECs). These cells, unlike the previous ones, do not show any cytoplasmic extension and have a rounded shape. Generally, SOM- and NPY-IR cells show a morphological appearance of “open type” EECs, while most GHR-IR cells show, on the contrary, the appearance of “closed type” EECs (Figure 2 and Figure 3). In addition, some NPY-IR cells were observed to co-express GHR and vice versa (Figure 3). Finally, these analyses surprisingly highlighted that the seabream gastric mucosa appears to have a much higher number of EECs.

### 3.4. Seabream Morphometric Results

From the evaluation of OPs-IR, it emerged that the integration of 5% and 10% of BP did not change the area occupied by OPs in the gastric mucosa of seabream. In detail: CTR 0.17 ± 0.02; BP 5% 0.14 ± 0.05; BP 10% 0.16 ± 004 (Figure 4A).

Regarding the number of EECs, we observed that the 10% BP group showed a statistically higher average number of NPY-IR cells (16.8 ± 7.5) compared with the CTR group (CTR 8.5 ± 4.8) (CTR vs. BP 10%; *p* ≤ 0.01), and we observed similarly with respect to the BP 5% group (10 ± 5.3) (BP 5% vs. BP 10% *p* ≤ 0.05) (Figure 4B). Likewise, compared to what we observed for the NPY-IR EECs, the EECs expressing SOM were also on average more numerous in the 10% BP diet (16.8 ± 3.5) and significantly different from the experimental CTR group (12.5 ± 5) (CTR vs. BP 10% *p* ≤ 0.05) and from the experimental group 5% (12.9 ± 2.5) (BP 5% vs. BP 10% *p* ≤ 0.01) (Figure 4C). No significant differences were observed regarding the mean number of GHR-IR cells in the three experimental groups (CTR 10.3 ± 3; BP 5% 9.8 ± 2.9 and BP 10% 12.4 ± 3.4, respectively) (Figure 4B). The percentage of colocalization, expressed as the number of NPY-IR cells that co-express GHR out of the total of GHR-IR cells, was 11.6% in the CTR group (84/723), 12% in the BP 5% group (84/702) and 13.6% in the BP 10% group (121/891).

### 3.5. Seabass Morphometric Results

As with the seabream, no statistical differences were observed in the seabass concerning the presence and distribution of OPs-IR within the three experimental groups. The following values were obtained: CTR 0.16 ± 0.02; BP 5% 0.17 ± 0.01; BP 10% 0.16 ± 0.01 (Figure 4D).

Unlike what was observed in the seabream, no significant differences were observed in the seabass gastric mucosa regarding the EECs expressing NPY (CTR 1.6 ± 0.4; BP 5% 1.4 ± 0.3; BP 10% 1.6 ± 0.4) and GHR (CTR 4.1 ± 0.7; BP 5% 4 ± 0.6; BP 10% 4.2 ± 0.6); it should be noted that there was only a slight increase in GHR-IR cells in the 10% BP diet (Figure 4E). We evaluated the percentage of cells co-expressing NPY/GHR on the total of EECs GHR-IR: (CTR 14.4%, 38/263; BP 5%, 28/289; BP 10%, 31/304).

EECs SOM-IR increased in the diet BP 10% (5.3 ± 0.7) compared to the BP 5% experimental group (4.4 ± 0.8) (BP 5% vs. BP 10% *p* ≤ 0.05) (Figure 4F).

## 4. Discussion

In aquaculture, fish protein hydrolysates have been evaluated as fishmeal replacers and attractants in several fish species [14,55,56,57,58,59]. Due to their good nutritional value and functional properties, protein hydrolysates derived from marine processing by-products have been considered an excellent ingredient in aquafeed [7], although, on the one hand, several authors report both beneficial effects [60,61,62] and no effect [11,63] of supplementation with protein hydrolysates. Previous studies have shown a significant improvement in the apparent digestibility of nutrients through the supplementation of dietary hydrolysates in the diets of red seabream [64] and seabass [65].

External factors, such as temperature, photoperiod, stress, predators, and availability of food, and internal factors, such as GI anatomy, species of fish, genetics, life phase, potential energy, etc., affected food intake and feeding behavior. In this context, the endocrine system (especially the enteroendocrine system) plays a key role in controlling appetite and, consequently, body weight in vertebrates. Most of the EECs dispersed in the mucosa of the GI tract can directly sense nutrients at their apical pole which faces the lumen of the gut and can be induced to release hormones into the circulation from their basal pole. Receptors of the G-protein-coupled receptor (GPCR) class have been implicated in EEC protein/amino acid sensing [66,67]. After enzymatic digestion of dietary proteins in the gut lumen, these EECs may be an easy target for functional peptides. Some EEC subtypes (e.g., GHR, SOM and enterochromaffin cells) do not contact the gut lumen [66]. In this context, free amino acids can also exert similar effects on EECs [67,68,69,70]. Some in vitro studies have shown that intact proteins and partially hydrolyzed proteins are more effective in inducing enteric hormone secretion than the products of their complete hydrolysis. Some selected synthetic peptides are more effective than the corresponding free amino acids [71,72,73].

The T1R1/T1R3 and T1R2/T1R3 heterodimers, the calcium-sensing receptor (CaSR) and the GPRC6A receptor can sense amino acids in the extracellular milieu and show partial selectivity for ligands [67]. In addition, in EECs in the stomach and small intestine, the GPR39 taste receptor is highly expressed. It is suggested that the GPR39 taste receptor is activated more by peptides than by free amino acids [66,74].

Both long-term and short-term pathways have been identified to regulate energy intake: long-term pathways inform the brain about body mass fluctuations, whereas short-term pathways relate to energy available in the GI. Both pathways use hormones, such as insulin and leptin, for long-term control and GHR, cholecystokinin and glucagon-like peptide-1 for short-term control [75].

It has been reported that the concentration of amino acids in the plasma is higher and rises more rapidly after the ingestion of peptide hydrolysates than after the ingestion of the corresponding whole proteins [76]. Moughan et al. [76] based their conclusions on the observation that solutions of peptide hydrolysates undergo a more rapid gastric emptying and intestinal absorption of their constituent amino acids than do the complete proteins from which they are derived. In this regard, oligopeptide carrier proteins (a large family of peptide transporters) play a key role in the uptake of dietary amino acids in di- and tripeptide forms in all vertebrates. This group of proteins includes PepT1, which shows high expression in the vertebrate intestine [77,78]. The *PepT1* gene has been described in Teleosts such as zebrafish *Danio rerio* and *Chionodraco hamatus* [79,80], cod (*Gadus morhua*) [81], and rainbow trout *(Oncorhynchus mykiss)* [82].

Furthermore, a functional correlation between PepT1 and some GI hormones, such as ghrelin, has been demonstrated in zebrafish [83] and grass carp (*Ctenopharyngodon idella*) [84]. Several authors suggest that intestinal nutrient transport mechanisms involving the intestinal peptide transporter PEPT1 are involved in the control of food intake in a high-protein diet through the involvement of the enteroendocrine system.

One of the most highly conserved neuropeptides in vertebrates [85,86] is NPY, a peptide of 36 amino acid residues first isolated from pig brain [87]. In some Teleosts, two classes of NPY (NPYa and NPYb) have been identified. However, teleost fish such as goldfish (*Carassius auratus*) and zebrafish (*Danio rerio*) have only NPYa [88,89]. NPY is known to be mainly secreted by the neurosecretory cells of the hypothalamus and is secreted in response to hunger [90,91]. Its primary function as a signaling factor is the regulation of a variety of biological processes such as food intake, the circadian rhythm, neuroendocrine functions and glucose homeostasis [92]. The *NPY* gene has been shown to be expressed in many tissues of several Teleosts. It is expressed in the central nervous system, intestine, liver, spleen, skeletal muscle and adipose tissue of several fish species, including zebrafish, goldfish (*Carassius auratus*), Atlantic salmon (*Salmo salar*), catfish (*Ictalurus punctatus*) and tilapia (*Oreochromis* sp.) [93,94,95,96]. In addition, fish NPY receptors are located in the brain, but can also be found in peripheral tissues, including the eye and intestine [97,98,99,100]. The GI tract is the largest peripheral NPY-producing organ, with distribution in all layers. In mammals, the distribution is greater in the submucosal and myenteric plexus neurons [101,102], whereas in fish, the greater distribution is observed in the EECs of the first intestine segments, as reported in pejerrey, *Odontesthes bonariensis* [103], R. quelen [104], dorado, *Salminus brasiliensis* [105], milkfish, *Chanos chanos* [106] and Nile tilapia (*O. niloticus*) [107]. In a fasting–refeeding study carried out in the blunt-nosed seabream (*M. amblycephala*), no differences were found in the expression of intestinal NPY, whereas the expression levels in the brain increased after fasting and decreased after refeeding [108]. These authors suggested that central NPY would act as an orexigenic factor. However, intestinal NPY would not act as a brain–gut peptide to stimulate appetite [109]. A high number of NPY cells in the gut may indicate that this region would act as a primary source of peripheral signals to stimulate feeding in the absence of food in this location, according to Vigliano et al. [103]. This suggests that the regulation of fish feeding is the result of the integration of different endocrine responses. Kondo et al. [110] observed how the integration of squid viscera hydrolysate in the red seabream (*Pagrus major*) diet significantly improved the expression of the brain NPY and intestinal GHR (as well as growth performance); the authors hypothesize that the positive eating behavior caused by squid viscera hydrolysate was controlled by these two hormones: NPY and GHR. There may be fish and species-specific mechanisms of agonism and antagonism [111,112]. It is reasonable to assume that the salmon-derived BPs tested in this trial stimulated, in the gastric mucosa, EECs expressing NPY differently in seabream than in seabass.

In this study, no differences were observed regarding the number of EECs GHR-IR in the experimental diets compared to the CTR. In overweight/obese individuals, Jensen et al. [113] hypothesized that supplementation with cod protein hydrolysate for 8 weeks would have a suppressive effect on postprandial GHR levels. These potentially beneficial effects were hypothesized to occur due to the presence of small peptides, mainly di- and tripeptides, which are rapidly absorbed from the GI tract and may affect appetite regulation pathways. The authors showed that fasting and postprandial concentrations of acylated GHR were not affected by 4 g of cod protein hydrolysate for 8 weeks. Jensen et al. [113] reported the effect of low-dose cod protein hydrolysate supplementation on fasting and postprandial acylated GHR levels: they observed no effect on fasting or postprandial acylated GHR levels. On the other hand, a single dose of 20 mg/kg body weight of cod protein hydrolysate had no effect on postprandial acylated GHR levels or hunger-related sensations in comparison with the control group [33]. Jönsson [30], in a review, reported that in tilapia and goldfish, GHR treatment appeared to increase food intake, whereas in rainbow trout, GHR decreased food intake, confirming that GHR affects (or does not affect) food intake by acting on well-established appetite signals in a species-specific manner. Finally, the increase in GHR observed in this study (although not statistically significant) is a positive aspect given the abundant evidence that GHR is an orexigenic peptide in several fish species, despite the data reported in the literature.

SOM has a primary role in the regulation of endocrine and exocrine secretion: it contributes to the reduction of gastric acid secretion and gastric motility and inhibits the secretion of several gastro-enteric and pancreatic hormones [114]. In the stomach, several regulatory peptides act in combination to regulate the secretion of acid and enzymes by the OPs [115,116] and the digestive processes in the small intestine, thereby optimizing the digestive functions of the entire GI tract.

Luminal application of glutamate in the rat stomach has previously been reported to increase the activity of the afferent fibers of the gastric branch of the vagus nerve by Uneyama et al. [117]. Interestingly, although each of the amino acids responded to luminal application in the gut [118,119], other amino acids did not show such effects. Thus, of the 20 dietary amino acids that are part of the body’s proteins, only glutamate can transmit nutritional information from the stomach to the brain. However, the exact mechanism for the luminal sensing of glutamate is still unclear [120].

Nakamura et al. [120] observed, in D (SOM-producing) cells obtained by fractionation of rat gastric cells, high expression of the CaSR, GPRC6A and Gi-coupling mGluRs receptors. Notably, CaSR expression was much higher on D cells than on parietal cells. The same authors also observed, in primary D cell cultures, that some specific CaSR amino acids (such as phenylalanine, tryptophan and histidine) significantly stimulated SOM release more than twofold, while other amino acid receptors (e.g., GPRC6A) favorable for lysine had no effect. They concluded that in the stimulation of SOM release in D cells, only the interaction between amino acids and specific receptors plays a functionally important role. Since SOM is an inhibitory regulator of gastric exocrine and endocrine secretion [121,122,123,124], it is reasonable to assume that inhibiting SOM release via amino acid coupled specific amino acid receptors (i.e., inhibiting the inhibitory effect) results in stimulating gastric secretion. This may partly explain the results we have obtained. In fact, in the gastric mucosa of seabream and seabass, we did not observe any changes of OPs in fish fed with 10% BP (and BP 5%) despite the increase in the number of SOM-IR EECs.

## 5. Conclusions

For the first time, the number and distribution of NPY, SOM and GHR-IR cells in the gastric mucosa of seabass and seabream have been shown to be affected by the administration of a BP-supplemented diet. Based on the data obtained, it is plausible to hypothesize that BPs perform, directly or indirectly, an action that stimulates the production of NPY, GHR and SOM without modifying the expression of OPs.

To gain more knowledge and to better understand the mechanisms of action of different bioactive peptides on the morphology of the gastric mucosa of fish, further studies are certainly needed.

In summary, the data suggest that salmon-derived BPs have a promising future as dietary ingredients for European seabass and seabream. This study also provides new insights into the morpho–functional changes that occur in the digestive tract of these species in response to feeding stimuli.

## Figures and Tables

**Figure 1 animals-13-03020-f001:**
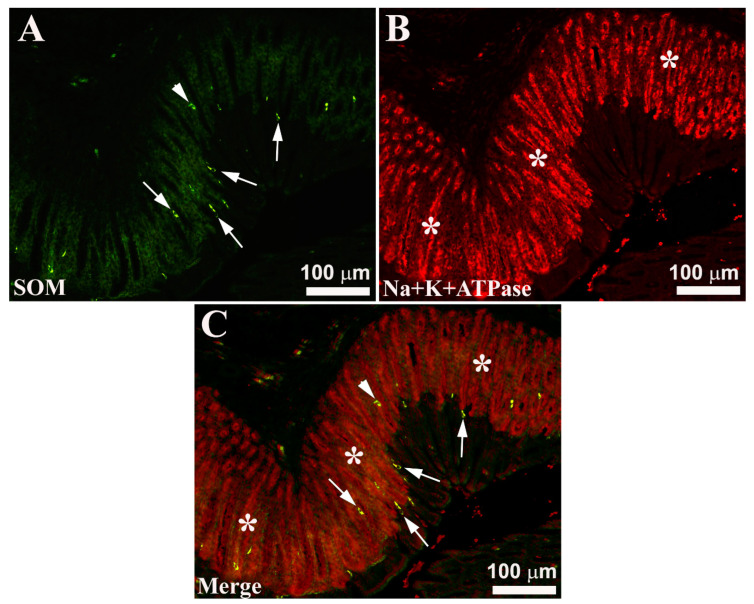
Seabream gastric mucosa. Image (**A**) shows the localization of immunoreactive (-IR) enteroendocrine cells (EECs) for somatostatin (SOM, arrows), while in (**B**) oxyntopeptic cells (asterisks) IR stained with Na^+^K^+^-ATPase antiserum. Occasionally, SOM-IR cells show a typical morphological appearance of ‘open type’ EECs ((**A**,**C**), arrows), while other SOM-IR cells show a ‘closed type’ morphological appearance ((**A**,**C**), arrowheads). Merge (**C**) shows the arrangement of SOM-IR cells in the gastric mucosa: most of the IR cells were positioned at the endoluminal side, while some SOM-IR cells are scattered along the adenomer of the gastric glands.

**Figure 2 animals-13-03020-f002:**
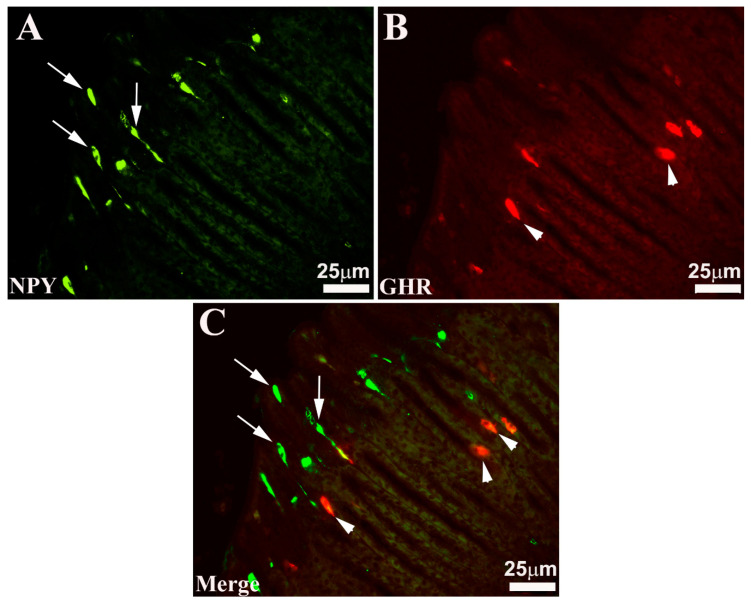
Seabass gastric mucosa. In these images, obtained at higher magnification, both the different morphological aspects and the different positions of the two subpopulations of immunoreactive (-IR) endocrine cells are represented. In the mucosa, neuropeptide Y (NPY, (**A**)) and ghrelin (GHR, (**B**)) -IR cells alternate which exhibit the two morphological types of “open type” ((**A**,**C**), arrows) and “closed type” enteroendocrine cells ((**B**,**C**), arrowheads). Merge (**C**) shows that not all enteroendocrine cells co-express NPY/GHR.

**Figure 3 animals-13-03020-f003:**
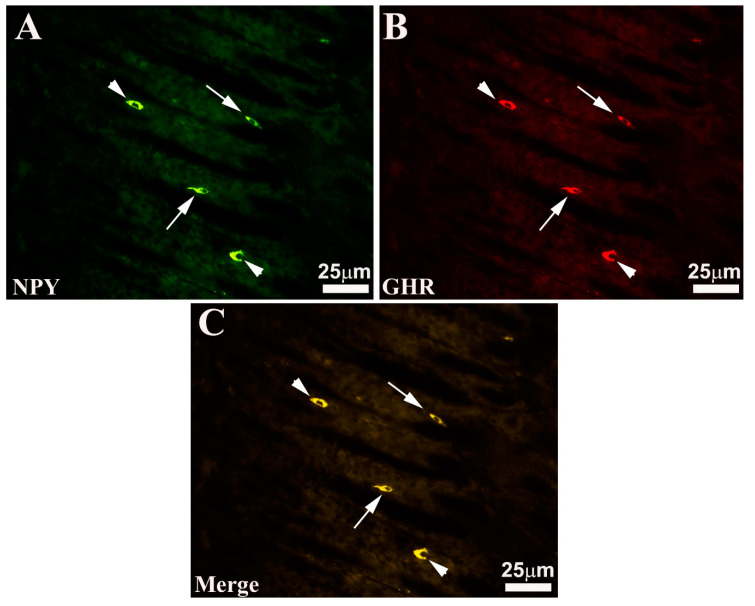
Seabass gastric mucosa. Images (**A**–**C**) show the total overlapping exhibited by the enteroendocrine neuropeptide Y (NPY) and ghrelin (GHR) immunoreactive (-IR) cells intermingled with oxyntopeptic cells. Note that some -IR enteroendocrine cells show the morphological appearance of “open type” ((**A**–**C**), arrows), while others of “closed type” ((**A**–**C**), arrowheads). (**C**) merge.

**Figure 4 animals-13-03020-f004:**
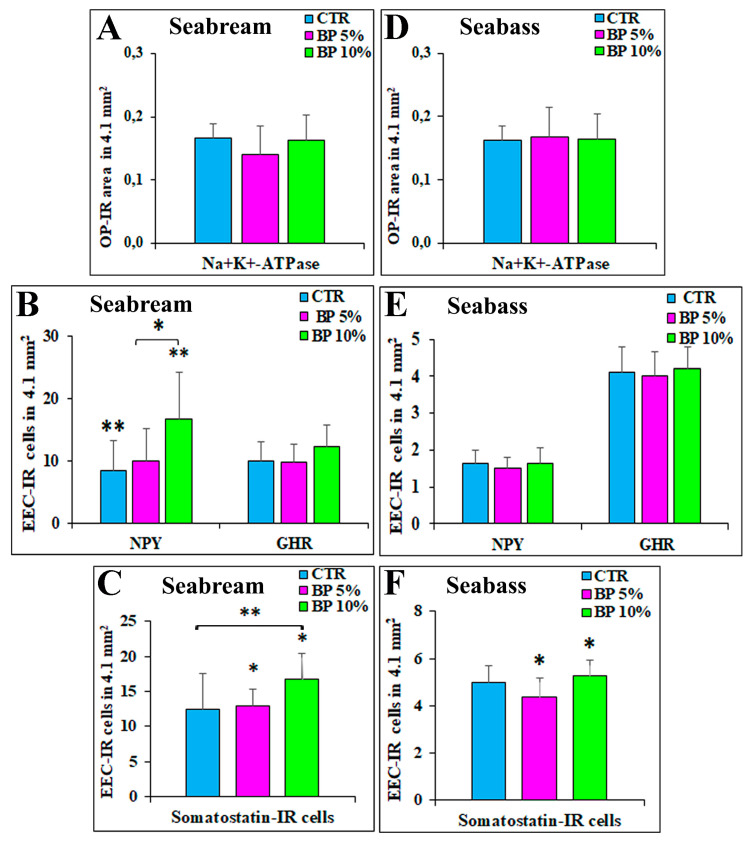
The histograms show the area occupied by oxyntopeptic (OP) immunoreactive (-IR) cells evaluated in the gastric mucosa of seabream (**A**) and seabass (**D**). (**B**) (seabream) and (**E**) (seabass) represent the mean number of neuropeptide Y (NPY) and ghrelin (GHR) -IR enteroendocrine cells (EECs), while E and F represent the mean number of somatostatin-IR cells in the seabream (**C**) and seabass (**F**) gastric mucosa. Experimental groups: CTR (control group), BP 5% and BP 10%. * indicates *p* ≤ 0.05; ** indicates *p* ≤ 0.01. Values expressed as mean + SD.

**Table 1 animals-13-03020-t001:** Components of the three experimental diets and their proximate composition.

	Experimental Diets		
	CTR	BP5	BP10
**Ingredients, %**			
Salmon hydrolysate	-	5.00	10.00
Fish meal	15.00	10.00	5.00
Soybean meal	15.00	15.00	15.00
Wheat	15.28	16.02	16.87
Wheat gluten	12.20	11.30	10.30
Corn gluten	8.00	8.00	8.00
Soy protein concentrate	5.40	5.40	5.40
Fish oil	10.00	9.90	9.80
Rapeseed oil	5.00	5.00	5.00
Horse beans	10.00	10.00	10.00
Lecithin from rapeseed	1.00	1.00	1.00
* Vitamin premix	0.50	0.50	0.50
* Mineral premix	0.50	0.50	0.50
Monosodiumphosphate	3.00	3.00	3.00
L-Lysine	0.40	0.40	0.40
DL-Methionin	0.05	0.05	0.05
**Proximate composition %**			
Moisture	6.41	6.46	6.57
Protein	39.90	39.50	39.22
Lipids	19.20	18.75	18.04
Ash	6.79	6.29	5.73

* Vitamin and mineral premix; (fulfilling recommendations for marine fish species given by NRC, 2011 [48]).

**Table 2 animals-13-03020-t002:** Chemical composition (g/100 g) of the spray-dried protein hydrolysate obtained from salmon raw material.

	Protein Hydrolysate
Crude protein	82.8
Water soluble protein	68.3
Total dry matter	98.3
Ash	6.3
Fat	11.2
Molecular weight (kDa) %	
>20	0.1
15–20	<0.1
10–15	0.1
8–10	0.2
6–8	0.9
4–6	3.9
2–4	16.5
1–2	24.3
0.5–1	19.4
0.2–0.5	13.9
<0.2	20.7

**Table 3 animals-13-03020-t003:** List and dilutions of primary and secondary antibodies.

Primary Antibodies	Code	Species	Dilution	Supplier
Ghrelin	AM26736PU-N	mouse	1:500	Acris
Somatostatin	ab16007	rat	1:500	Enzo Life Sciences
Neuropeptide Y	NBP1-46535	goat	1:1000	Novus Biological
Na^+^K^+^-ATPase	GLP-1(1-36) # 9153	rabbit	1:600	Abcam
**Secondary antibodies**			**Dilution**	**Supplier**
goat anti-rat FITC			1:1000	Proteintech^®^
goat anti-rabbit Alexa Fluor^®^ 594			1:1000	Thermofisher/Invitrogen
donkey anti-mouse Alexa Fluor^®^ 594			1:1000	Thermofisher/Invitrogen
donkey anti-goat Alexa Fluor^®^ 488			1:1200	Thermofisher/Invitrogen

Acris Antibodies GmbH OriGene Company, Schillerstraße 5, Herford, Germany. Enzo Life Sciences, New York, NY, USA. Novus Biological, Centennial, CO, USA. Abcam, Cambridge, UK. Proteintech^®^ Group Inc, Rosemont, IL, USA. Thermo Fisher/Invitrogen Scientific, Waltham, MA, USA.

## Data Availability

All the datasets used and/or analyzed throughout the present study are available from the corresponding author on reasonable request.

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
