# Peer review of "Effects of Bioactive Peptides from Atlantic Salmon Processing By-Products on Oxyntopeptic and Enteroendocrine Cells of the Gastric Mucosa of European Seabass and Gilthead Seabream"

_animals, 2023, doi:10.3390/ani13193020_

Round 1
Reviewer 1 Report
The comments to be sent to the authors are in the attached file.
I remain at your disposal for any clarifications you may need.

Author Response
Reviewer 1
Although the manuscript presents an interesting subject for fish farming, my opinion is NOT to publish the article as presented. The article must be rewritten and submitted for further evaluation. The reasons that led me to this rejection, and must be justified in the new manuscript, were:
- Non-performance of physiological parameters: Although, in the last decade, many studies on the neuroendocrine-SN system and its peptides in fish have been carried out, further studies are still needed to investigate the physiological role of these peptides in the gastrointestinal system, Parameters how the stimulation of these bioactive peptides on digestive enzymes, on gastrointestinal tract-GT motility or on another parameter that alter digestive physiology should be investigated since there was no effect on fish consumption or performance.
We thank the reviewer for his/her comment: The topic of this study was to carry out a morphological-morphometric evaluation of oxyntopeptic cells and a subpopulation of enteroendocrine cells (neuropeptide Y, ghrelin and somatostatin endocrine cells). As the reviewer will no doubt have ascertained, the bibliography pertaining to the qualitative/quantitative study of oxyntopeptic cells as well as certain subpopulations of enteroendocrine cells in the gastric mucosa of fish is practically absent. Having said this, I agree with the reviewer that physiological studies on the effects of bioactive peptides on the endocrine system would have enhanced the manuscript. This study was a preliminary part, certainly in the future we will also evaluate other parameters such as (as suggested) digestive enzymes, intestinal motility, hormone blood level etc. Finally, it was shown in this study that although the number of certain endocrine cell subpopulations was affected, these bioactive peptides had no effect on food consumption and growth performance. Even this result, which was not expected, is nevertheless one that fellow researchers may take into consideration in the future when formulating diets supplemented with bioactive peptides at the rates used in this study and for the species considered (sea bream and sea bass, respectively).
- Collection of biological samples for NS: Why was the collection performed exclusively in the stomach? The presence of SN is different in the different segments of the IG and also between species. Generally, manuscripts about SN present a detailed description of the GT and this can generate some doubts, for example, justifying the same fasting period for two species, are the GT similar???
We thank the reviewer for his/her comment. We decided to consider the stomach as a topic because, as mentioned above, the gastric compartment of fish is little/poorly studied. We also did preliminary studies (unpublished) on the cranial, middle and caudal intestine and evaluated NPY, ghrelin and somatostatin immunoreactive cells. The number of these cells was very low/scarce and, in some intestinal tracts, we detected no immunoreactivity.
- Discussion: In this topic, the results were not discussed, a literature review was carried out and even presented manuscripts with response parameters that were not evaluated in the study. I suggest that the discussion addresses the differences found in other manuscripts, such as the influence due to the anatomy of the TG, eating habits, etc..., especially in the case of studies with two species.
Thanks to the reviewer for the comment. In the discussions section we have added/integrated some parts where our results are compared with the results obtained by other colleagues.
- Although some of the articles that I suggest reading have been presented in the reference, I suggest some papers for reading and basis for the discussion:
Li, Peng, Mai, Kangsen, Trushenski, Jesse, & Wu, Guoyao. (2009). New developments in fish amino acid nutrition: towards functional and environmentally oriented aquafeeds. Amino Acids, 37(1), pp. 43−53. doi: 10.1007/s00726-008-0171-1
Shamushaki VAJ, Kasumyan AO, Abedian A, Abtahi B (2007) Behavioural responses of the Persian sturgeon (Acipenser persicus) juveniles to free amino acid solutions. Mar Fresh Behav Physiol 40:219–224
Distribution of goblet and endocrine cells in the intestine: A comparative study in Amazonian freshwater Tambaqui and hybrid catfish. Journal of Morphology. https://doi.org/10.1002/jmor.21079
Endocrine cells producing peptide hormones in the intestine of Nile tilapia: distribution and effects of feeding and fasting on the cell density. Fish Physiology and Biochemistry. https://doi.org/10.1007/s10695-017-0380-1
Relative distribution of gastrin- CCK-8-, NPY- and CGRP-immunoreactive cells in the digestive tract of dorado (Salminus brasiliensis). Tissue & Cell, v. 47, p. 123-131, 2015.
Feed intake and gene expression of appetite-regulating hormones in Salminus brasiliensis fed diets containing soy protein concentrate. Comp. Biochem. Physiol. A. 268, 111208.
Growth performance, health, and gene expression of appetite-regulating hormones in Dourado Salminus brasiliensis, fed vegetable-based diets supplemented with swine liver hydrolysate. Aquaculture, 548(2): 737640.
Appetite-controlling endocrine systems in teleosts. Front. Endocrinol, (2017), 8:73. doi.org/10.3389/fendo.2017.00073.
We thank the reviewer for his/her comment. In addition to the recommended articles, we have integrated and updated with recent publications.
Reviewer 2 Report
The manuscript is well-written and treats an interesting topic describing for the first time the effects of fish protein hydrolysate on gastric mucosa oxyntopeptic and enteroendocrine cells of seabass and seabream. The authors evaluated both cell distribution and the expression of gut-brain hormones.
The study presentation is comprehensive and detailed but there are some minor points that should be clarified:
The number of fish used contradicts itself in the manuscript, since in the abstract at line 24, it is indicated that 27 fish/species and in materials and methods at line 150 are reported to be 60 fish/species. Please describe better the experimental design.
Since GHR, SOM, and NPY are involved in the regulation of appetite, the author should indicate whether fish were postprandial or fasted sampled in the Materials and Methods section. This could help interpret the results.
Since hydrolyzed proteins are rich in di- and tri-peptides, it would have been interesting to evaluate the expression of intestinal peptide transporters, such as PEPT1, whose expression is regulated by ghrelin and leptin. The authors have just hinted at the role of intestinal absorption in the discussion (lines 364-367). In my opinion, this aspect should be considered thoroughly.
Minor revisions:
Materials and methods: line 110 species name “Salmo salar” must be written in latin
Results: In Table 4, FCR data have not been included, but in the materials and methods, the authors listed it among the calculated growth parameters.
In Figure 4 change “sea bream” and “sea bass” with “seabream” and “seabass”, respectively
Author Response
Reviewer 2
The manuscript is well-written and treats an interesting topic describing for the first time the effects of fish protein hydrolysate on gastric mucosa oxyntopeptic and enteroendocrine cells of seabass and seabream. The authors evaluated both cell distribution and the expression of gut-brain hormones.
The study presentation is comprehensive and detailed but there are some minor points that should be clarified:
The number of fish used contradicts itself in the manuscript, since in the abstract at line 24, it is indicated that 27 fish/species and in materials and methods at line 150 are reported to be 60 fish/species. Please describe better the experimental design.
We thank the reviewer for his/her comment. We have modified the material and methods part to clarify what the reviewer requested.
Since GHR, SOM, and NPY are involved in the regulation of appetite, the author should indicate whether fish were postprandial or fasted sampled in the Materials and Methods section. This could help interpret the results.
We thank the reviewer for his/her comment. Thanks to the reviewer for the comment. We have added what was requested in the materials and methods section.
Since hydrolyzed proteins are rich in di- and tri-peptides, it would have been interesting to evaluate the expression of intestinal peptide transporters, such as PEPT1, whose expression is regulated by ghrelin and leptin. The authors have just hinted at the role of intestinal absorption in the discussion (lines 364-367). In my opinion, this aspect should be considered thoroughly.
I thank the reviewer for the very intriguing comment. We have integrated the manuscript by describing the PepT1 peptide transporter, its presence in fish and its regulation with some hormones such as ghrelin.
Minor revisions:
Materials and methods: line 110 species name “Salmo salar” must be written in latin
Ok, done.
Results: In Table 4, FCR data have not been included, but in the materials and methods, the authors listed it among the calculated growth parameters.
Thanks to the reviewer for the comment. Regard to the table 4, in agreement with colleagues (co-authors) we decided to eliminate table 4 because the manuscript is based on the anatomical description of the stomach of sea bass and sea bream and the previously reported values were not then discussed. We have briefly reported some parameters which show that there were no differences and the different ranges of values obtained (FBW, FI, SGR etc.).
In Figure 4 change “sea bream” and “sea bass” with “seabream” and “seabass”, respectively
Ok, done.
